# Logical Credal Networks

**Radu Marinescu**
IBM Research
radu.marinescu@ie.ibm.com

**Haifeng Qian**
AWS AI Labs
qianhf@amazon.com

**Alexander Gray**
IBM Research
alexander.gray@ibm.com

**Debarun Bhattacharjya**
IBM Research
debarunb@us.ibm.com

**Francisco Barahona**
IBM Research
barahon@us.ibm.com

**Tian Gao**
IBM Research
tgao@us.ibm.com

**Ryan Riegel**
IBM Research
ryan.riegel@ibm.com

**Pravinda Sahu**
IBM Consulting
pravisah@in.ibm.com

## Abstract

We introduce Logical Credal Networks (or LCNs for short) – an expressive probabilistic logic that generalizes prior formalisms that combine logic and probability. Given imprecise information represented by probability bounds and conditional probability bounds on logic formulas, an LCN specifies a set of probability distributions over all its interpretations. Our approach allows propositional and first-order logic formulas with few restrictions, e.g., without requiring acyclicity. We also define a generalized Markov condition that allows us to identify implicit independence relations between atomic formulas. We evaluate our method on benchmark problems such as random networks, Mastermind games with uncertainty and credit card fraud detection. Our results show that the LCN outperforms existing approaches; its advantage lies in aggregating multiple sources of imprecise information.

## 1 Introduction

Graphical models provide a powerful framework for reasoning about conditional dependency structures over many variables. Bayesian networks [26, 19], for example, rely on precise probabilistic information and, therefore, encode a single distribution over the variables of interest, while credal networks and their variants [7, 3, 8, 9], which encode sets of probability distributions, are able to reason effectively with imprecise information represented by sets of probability values or intervals.

Many (if not all) real-world applications require efficient handling of uncertainty and a compact representation of a wide variety of knowledge. Indeed, complex concepts and relationships that typically comprise expert knowledge may be difficult to express in graphical models but can be represented compactly using classical logic. Consequently, probabilistic logic which combines probability and logic in a principled manner has emerged over the years as a unified framework to deal effectively with these complex domains [23, 12, 16, 17, 1, 5, 11, 29, 15, 27, 28]. While some of these formalisms (e.g., [29, 15, 27]) associate a single real value to the logical formulas to represent the uncertainty around their truth values, others (e.g., [23, 12, 24]) relax this requirement and allow specifying lower and upper probability bounds on logical formulas.

In practice, knowledge is typically imprecise and it is often the case that multiple sources of knowledge need to be combined effectively to improve the solution quality of the problem at hand. For example, in a practical credit card fraud detection application, it would be desirable to combine a statistical model

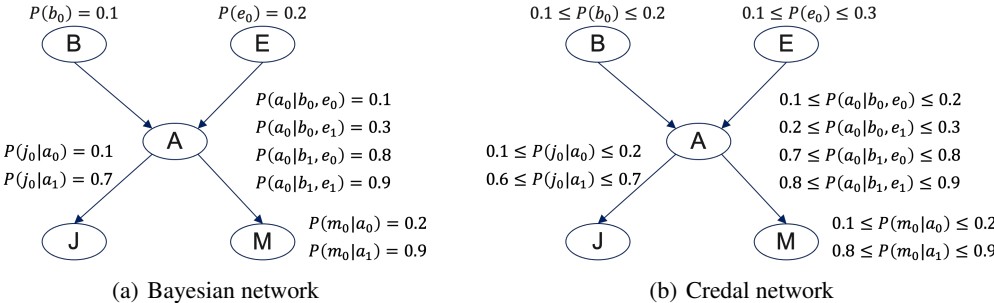

(a) Bayesian network
(b) Credal network

Figure 1: Examples of simple Bayesian and credal networks.

capturing the uncertainty around historical transaction data with probabilistic logic rules expressing imprecise expert knowledge about the domain in order to predict future fraudulent transactions more accurately. Existing formalisms such as [23, 12, 24] allow bounds on logic formulas to model imprecise knowledge. They also impose few restrictions on logic formulas for increased expressivity but lack independence declarations which typically leads to excessively wide posterior probability intervals in the inference results and that may not be very useful for the decision maker.

**Contribution:** In this paper, we present *Logical Credal Networks* (LCN), a new probabilistic logic model that allows probability bounds and conditional probability bounds on arbitrary propositional and first-order logic formulas without requiring acyclicity and other strong restrictions. In addition, we define a generalized Markov condition for LCNs that allows us to make some independence assumptions between the atoms occurring in the logic formulas. Specifically, this allows atomic formulas in an LCN to be treated as independent unless there is information saying otherwise. The independence assumptions in LCNs mirror similar assumptions present in probabilistic databases [4] where for example tuples are considered independent of each other unless certain constraints that encode correlation are violated [14, 30, 20]. Furthermore, we show that even though LCNs allow cyclic dependencies between variables, our proposed Markov condition matches the Markov condition in Bayesian and credal networks for acyclic graphs. Subsequently, we describe an exact inference algorithm to compute lower and upper bounds on the posterior probability of a query formula (optionally, in the presence of evidence). We experiment with random LCNs as well as with benchmark problems derived from Mastermind games with uncertainty and a realistic credit card fraud detection application. Our results are quite promising and show that LCNs outperform significantly existing approaches, especially in terms of solution quality. In particular, the results highlight the ability of LCNs to effectively aggregate multiple sources of imprecise information.

The supplementary material includes additional technical details and examples.

## 2 Background

### 2.1 Bayesian and Credal Networks

A *Bayesian network* (BN) [26] is defined by a tuple $\langle \mathbf{X}, \mathbf{D}, \mathbf{P}, G \rangle$, where $\mathbf{X} = \{X_1, \ldots, X_n\}$ is a set of variables over multi-valued domains $\mathbf{D} = \{D_1, \ldots, D_n\}$, $G$ is a directed acyclic graph (DAG) over $\mathbf{X}$ as nodes, and $\mathbf{P} = \{P_i\}$ where $P_i = P(X_i | pa(X_i))$ are *conditional probability distributions* (CPDs) associated with each variable $X_i$ and its parents $pa(X_i)$ in $G$. The Bayesian network represents a joint probability over $\mathbf{X}$, $P(X_1, \ldots, X_n) = \prod_{i=1}^{n} P(X_i | pa(X_i))$.

*Credal networks* (CN) [6, 3] extend Bayesian networks to deal with imprecise probabilities. More specifically, each variable $X_i$ and each configuration $\pi_{ik}$ of its parents $pa(X_i)$ in a credal network is associated with a *conditional credal set* $K(X_i | pa(X_i) = \pi_{ik})$ which is specified separately from all others and is assumed to be closed and convex (e.g., probability intervals). The *strong extension* of a credal network is the convex hull of all joint distributions that satisfy the property that every variable is strongly independent of its non-descendants conditional on its parents [6].

**Example 1.** *Figures 1(a) and 1(b) show simple examples of Bayesian and credal networks defined over five binary variables together with their corresponding conditional probability tables and credal*

*sets, respectively. The lowercase letters denote the domain values of the variables (e.g., $A \in \{a_0, a_1\}$). In this case, the credal sets are given by closed probability intervals (e.g., $P(b_0) \in [0.1, 0.2]$).*

## 2.2 Probabilistic Logics

Propositions are denoted by lowercase letters $x, y, z, \ldots$ and propositional literals (i.e., $x$, $\neg x$) stand for $x$ being $True$ or $x$ being $False$. A *term* is a variable, a constant, or a function applied to terms. An *atom* (or atomic formula) is either a proposition or a predicate $p(t_1, \ldots, t_k)$ of arity $k$ where the $t_i$ are terms. A *formula* is built out of atoms using logical connectives $\neg, \vee, \wedge$ and $\rightarrow$, respectively. For simplicity, we assume that first-order logic (FOL) formulas are universally quantified.

A probabilistic logic is defined by a set of logic formulas[1] $(q, \alpha_q)$ such that each formula $q$ is annotated with a probability value $\alpha_q \in [0, 1]$ representing the probability $P(q)$ of $q$ being true [23]. The semantics of this logic is the set of probability distributions over all interpretations such that $P(q) = \alpha_q$[2]. The $\alpha_q$ values can be replaced with intervals to deal with imprecise probabilities. Given a query formula $f$, a common inference task is to compute lower and upper probability bounds on $P(f)$, denoted by $\underline{P}(f)$ and $\overline{P}(f)$, respectively.

A major weakness of this probabilistic logic [23] (as well as more recent work on description logic [16, 17, 22]) is the lack of a Markov condition: there are no independence relations that are implied by the probabilistic logic. Consider the following simple example:

$$0.3 \leq P(x) \leq 0.7 \qquad\qquad 0.3 \leq P(y) \leq 0.7 \qquad\qquad (1)$$

where $x, y$ are atomic formulas. Computing the bounds on $P(x \oplus y)$[3] results in the interval $[0, 1]$ [23]. Indeed, there exists a joint distribution over $x, y$ such that (1) is satisfied and that $x \oplus y$ is always $False$, and there exists another such that $x \oplus y$ is always $True$. The inference result of $[0, 1]$, however, is not informative for most purposes and often is not the intention when one writes down (1) for an application. A more practical approach in this case is to assume that $x$ and $y$ are independent of each other unless there is information saying otherwise. With this independence assumption, solving the original query $P(x \oplus y)$ results in the interval $[0.42, 0.58]$.

One way to allow for a Markov condition in probabilistic logic is to constrain the logic formulas to a specific structure such as a Bayesian network [1] or a credal network with probability intervals [8, 9]. However, this representation comes with several restrictions: (1) the only non-atomic logic formulas allowed are AND over atomic formulas and negation of atomic formulas; (2) there must not be cyclic dependencies among atomic formulas; (3) an atomic formula must be specified by either a marginal probability interval or by a set of conditional probability intervals, and not both; (4) the conditions in the conditional probabilities must enumerate all possible interpretations of the parent variables – we will refer to the last two requirements as the *unique-assessment assumption*. In practice, there is often knowledge that cannot be expressed by a simple AND. Furthermore, there could be multiple sources of information that, when aggregated, break the acyclicity or unique-assessment requirements.

# 3 Logical Credal Networks

In this section, we introduce the *Logical Credal Network* (LCN) – a new probabilistic logic designed to allow as few restrictions as possible on logic formulas when specifying probability bounds together with a set of implied independence relations similar to those present in Bayesian and credal networks.

## 3.1 Syntax

An LCN is defined by a set of *probability sentences* having one of the following two forms:

$$l_q \leq P(q) \leq u_q \qquad\qquad (2)$$
$$l_{q|r} \leq P(q \mid r) \leq u_{q|r} \qquad\qquad (3)$$

where $q$ and $r$ can be arbitrary propositional or FOL formulas and $0 \leq l_q \leq u_q \leq 1$, $0 \leq l_{q|r} \leq u_{q|r} \leq 1$. Each sentence is further associated with a Boolean parameter $\tau \in \{True, False\}$, which indicates whether a sentence implies dependence between the atomic formulas occurring in $q$.

---

[1] A logic formula $q$ can be either a propositional formula or a universally quantified first-order logic formula.
[2] We use $P(q)$ as shorthand notation for $P(q$ is True$)$ and $P(q \mid r)$ for $P(q$ is True $\mid r$ is True$)$.
[3] Symbol $\oplus$ stands for the logical XOR operator.

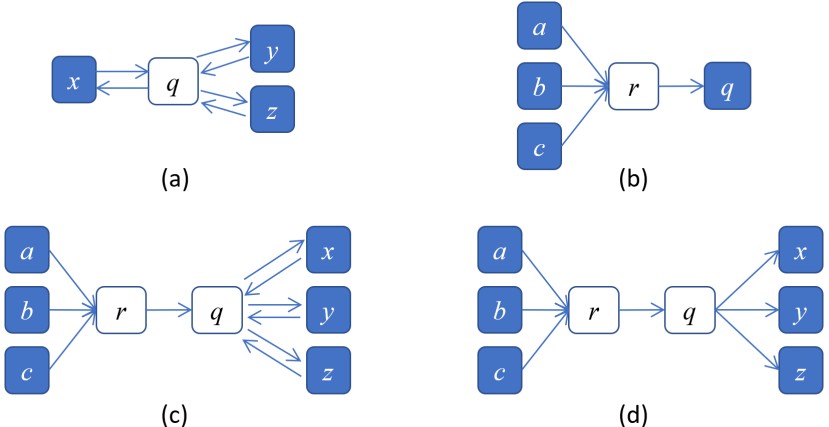

Figure 2: Stamps of the sentences in LCNs: (a) A sentence (2) with $\tau = True$; (b) A sentence (3) where $q$ is an atomic formula ($\tau$ is either $True$ or $False$); (c) A sentence (3) where $q$ is a non-atomic formula and $\tau = True$; (d) A sentence (3) where $q$ is a non-atomic formula and $\tau = False$. The nodes corresponding to the atomic formulas are shaded.

**Example 2.** *Consider the following Smokers and Friends LCN (adapted from [29]). The predicates $Fr(\cdot, \cdot)$, $Sm(\cdot)$ and $Ca(\cdot)$ stand for $Friends(\cdot, \cdot)$, $Smokes(\cdot)$ and $Cancer(\cdot)$, respectively. Furthermore, predicate $Fr(\cdot, \cdot)$ is symmetric.*

$$0.5 \leq P\left(Fr\left(\alpha, \gamma\right) \mid Fr\left(\alpha, \beta\right) \wedge Fr\left(\beta, \gamma\right)\right) \leq 1, \tag{4}$$

$$0 \leq P\left(Sm\left(\alpha\right) \oplus Sm\left(\beta\right) \mid Fr\left(\alpha, \beta\right)\right) \leq 0.2, \tag{5}$$

$$0.03 \leq P\left(Ca\left(\alpha\right) \mid Sm\left(\alpha\right)\right) \leq 0.04, \tag{6}$$

$$0 \leq P\left(Ca\left(\alpha\right) \mid \neg Sm\left(\alpha\right)\right) \leq 0.01, \tag{7}$$

*The LCN sentences state the following: friends of friends are likely friends (4); if two people are friends, they likely either both smoke or neither does (5); smoking likely causes cancer (6) (7).*

## 3.2 Semantics

An LCN represents the set of all its models. A *model*[4] of an LCN is a probability distribution over all interpretations such that it satisfies a set of constraints given explicitly by (2)(3) and a set of independence constraints which are implied by the LCN. The latter involve atomic formulas only and are similar to the independence relations implied by a Markov condition in graphical models [19]. Furthermore, we say that an LCN is *consistent* if it has at least one model. Otherwise, it is *inconsistent*.

In contrast to previous work [1, 6, 9], we propose a generalized Markov condition that accommodates the LCN's much more relaxed requirements on logic formulas, including cyclic dependencies (i.e., specifying sentences involving $P(x|y)$ and $P(y|x)$ is allowed in LCNs because they may come from different sources of information). We introduce next the *stamp* of an LCN sentence and, subsequently, define the *primal graph* of an LCN which is the basis of our proposed Markov condition.

**Definition 1** (stamp). *Given an LCN $\mathcal{L}$ and a sentence $s \in \mathcal{L}$, the stamp $G(s)$ of $s$ is a directed graph whose nodes correspond to formulas in $s$, together with all atoms occurring in formulas in $s$. The edges of $G(s)$ are determined as follows. For $\tau = True$, (i) if $s$ is (2) such that $q$ has atoms $x_1, \ldots, x_n$ then $G(s)$ has directed edges from $q$ to each $x_i$, and from each $x_i$ to $q$, respectively, and (ii) if $s$ is (3) such that $x_1, \ldots, x_n$ (resp. $a_1, \ldots, a_m$) are the atoms in $q$ (resp. $r$), then $G(s)$ contains a directed edge from $r$ to $q$, a set of directed edges from each $a_j$ to $r$, as well as directed edges from $q$ to each $x_i$ and from each $x_i$ to $q$, respectively. For $\tau = False$, (iii) if $s$ is (2) then $G(s)$ has no edges, and (iv) if $s$ is (3) such that $q$ and $r$ have atoms $x_1, \ldots, x_n$ and $a_1, \ldots, a_m$ then $G(s)$ contains directed edges from each $a_j$ to $r$, from $r$ to $q$ and from $q$ to each $x_i$, respectively.*

---

[4]The semantics is not model-theoretic because there exist implied constraints that are jointly derived from multiple sentences.

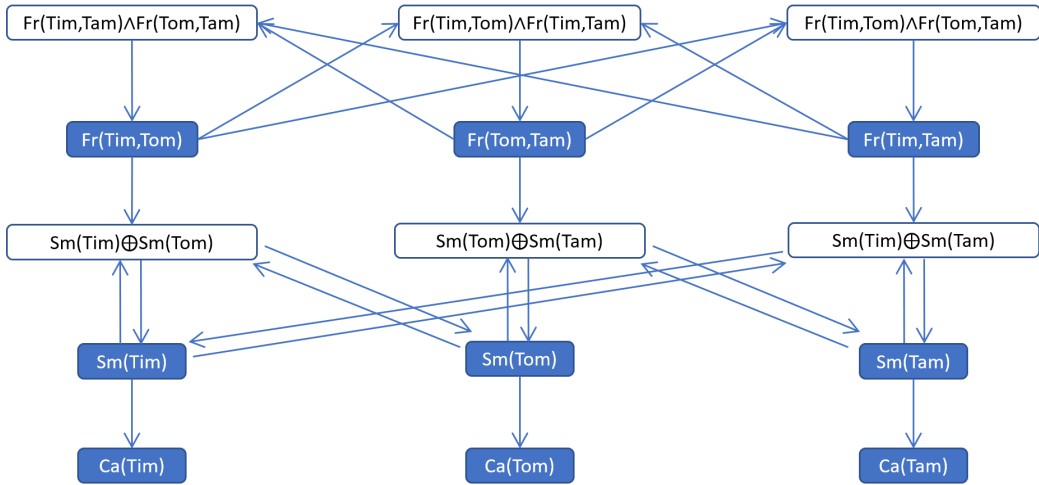

Figure 3: The primal graph of the LCN from Example 2 grounded on the domain $\{Tim, Tam, Tom\}$.

Figure 2 shows the stamps corresponding to different sentences in LCNs. The intuition behind the stamps is the need to capture two types of dependencies. For a sentence (2) with $\tau = True$, the dependency among atomic formulas in $q$ is similar to a clique in Markov networks (see Figure 2(a)). For a sentence (3) where $q$ is an atomic formula, the dependency is similar to the dependencies in Bayesian networks (see Figure 2(b)). The stamp of Figure 2(c) is a composition of the two types. For a sentence where $q$ is non-atomic, the parameter $\tau$ controls whether its stamp includes edges from its atoms $x$, $y$ and $z$ to $q$, and consequently modifies the primal graph and the Markov condition.

**Definition 2** (primal graph). *The* primal graph *of an LCN $\mathcal{L}$ with $n$ sentences $s_1, \ldots, s_n$ is the directed graph $G$ representing the union of the stamps associated with the sentences in $\mathcal{L}$, namely $G = G(s_1) \cup \cdots \cup G(s_n)$.*

Figure 3 illustrates the primal graph of the LCN from Example 2 grounded on a domain of three people $x \in \{Tim, Tam, Tom\}$, where we assume $\tau = True$ for all sentences. As before, the symbol $\oplus$ is XOR and the shaded nodes correspond to atomic formulas.

**Definition 3** (parents). *The* parents *of an atomic formula $x$, denoted by $parents(x)$, is the set of atomic formulas $y$ such that there exists a directed path $(y \to z_1 \to \cdots \to z_k \to x)$ from $y$ to $x$ in the primal graph where all intermediate nodes $z_i$ (if any) are non-atomic.*

**Definition 4** (descendants). *The* descendants *of an atomic formula $x$, denoted by $descendants(x)$, is the set of atomic formulas $y$ such that there exists a directed path $(x \to z_1 \to \cdots \to z_k \to y)$ from $x$ to $y$ in the primal graph where none of the intermediate nodes $z_i$ (if any) is in $parents(x)$.*

**Definition 5** (Markov condition). *Let $\mathcal{L}$ be an LCN and $M$ be a model of $\mathcal{L}$. Given $M$, every atomic formula $x$ is conditionally independent of its non-descendant non-parent variables $ndnp(x)$ given $parents(x)$, where $ndnp(x) \triangleq \{atomic\ formulas\} \setminus \{parents(x) \cup descendants(x) \cup \{x\}\}$.*

Clearly, the generalized Markov condition allows us to make additional independence assumptions by inspecting the primal graph of the LCN. For example, looking at Figure 3 again, we see that $parents(Sm(Tim)) = \{Fr(Tim, Tom), Fr(Tim, Tam), Sm(Tom), Sm(Tam)\}$ and $ndnp(Sm(Tim)) = \{Fr(Tom, Tam), Ca(Tom), Ca(Tam)\}$. Therefore, $Sm(Tim)$ is conditionally independent of $Fr(Tom, Tam)$, $Ca(Tom)$ and $Ca(Tam)$ given $Fr(Tim, Tom)$, $Fr(Tim, Tam)$, $Sm(Tom)$ and $Sm(Tam)$. Similarly, we can determine that $Ca(Tim)$ is conditionally independent of all other variables given $Sm(Tim)$.

**Remark** Let us consider the case when an LCN is used to represent a Bayesian network $\mathcal{B}$. In this case, the upper and lower bounds in sentences (2)(3) are equal; formulas involved in (2)(3) are propositional and atomic; and the LCN sentences specify the prior and conditional probabilities in the Bayesian network. The primal graph is constructed by stamps only in the form of Figure 2(b) and is identical with the DAG of $\mathcal{B}$. Consequently, the parents (resp. descendants) of an atomic node $x$ are the same as the parents (resp. descendants) of $x$ in $\mathcal{B}$. Therefore, the Markov condition of the

LCN is identical to that defined for Bayesian networks [26]. The LCN has only one model, which is the same probability distribution as the one represented by the Bayesian network $\mathcal{B}$.

Similarly, if an LCN is composed of sentences (2) only and $\tau = True$ for each sentence then, by our definitions, the parents of an atomic node $x$ are the same as the Markov blanket of $x$ in a Markov Logic Network (MLN) with the same logic formulas as the LCN; the descendants of any atomic node $x$ is always empty; ndnp $(x)$ are simply the atomic nodes not in the Markov blanket. Therefore, our Markov condition in this case is identical to the independence relations encoded in the MLN [29].

### 3.3 Inference

Given an LCN $\mathcal{L}$, a query formula $f$ and, optionally, some evidence $E = \{e_1, \ldots, e_t\}$ (i.e., a subset of atomic formulas that are true), *marginal inference* in LCNs calls for computing lower and upper bounds on the posterior marginal probability $P(f|E)$, denoted by $\underline{P}(f|E)$ and $\overline{P}(f|E)$, respectively.

The task entails solving a non-linear program defined over a set of variables representing the probabilities of $\mathcal{L}$'s interpretations and comprising of linear constraints derived from the LCN sentences as well as non-linear constraints corresponding to the independence assumptions derived from $\mathcal{L}$'s Markov condition, respectively. The query $P(f|E)$ is translated into a non-linear objective function which is subsequently minimized and maximized, thus yielding $\underline{P}(f|E)$ and $\overline{P}(f|E)$, respectively.

More specifically, if $\mathcal{L}$ has $n$ atomic formulas then there are $N = 2^n$ possible interpretations (recall that in the case of FOL, $\mathcal{L}$ refers to its grounding on the domains of its variables as shown in the example of Figure 3). Let $\vec{p} = (p_1, \ldots, p_N)$ be the vector of their probabilities and let $\vec{A}_\alpha = (a_1^\alpha, \ldots, a_N^\alpha)$ be a binary vector, called an *indicator vector*, such that $a_j^\alpha$ is 1 if formula $\alpha$ is true in the $j$-th interpretation and 0 otherwise. Firstly, $\vec{p}$ must be a valid probability distribution:

$$\sum_{j=1}^{N} p_j = 1 \quad \text{and} \quad p_j \geq 0, \forall j = 1, \ldots, N \tag{8}$$

Since the probability of a formula is the sum of the probabilities of the interpretations in which the formula is true, sentences (2) and (3) in $\mathcal{L}$ are encoded by constraints (9) and (10), respectively:

$$\vec{A}_q \odot \vec{p} \geq l_q \quad \text{and} \quad \vec{A}_q \odot \vec{p} \leq u_q \tag{9}$$

$$\vec{A}_{q \wedge r} \odot \vec{p} - l_{q|r} \cdot \vec{A}_r \odot \vec{p} \geq 0 \quad \text{and} \quad \vec{A}_{q \wedge r} \odot \vec{p} - u_{q|r} \cdot \vec{A}_r \odot \vec{p} \leq 0 \tag{10}$$

where $\odot$ is the dot product of two vectors (i.e., $\vec{a} \odot \vec{b} = \sum_{j=1}^{n} a_j \cdot b_j$), $\vec{A}_q$ and $\vec{A}_{q \wedge r}$ are the indicator vectors for the interpretations where formulas $q$ and $q \wedge r$ are true, respectively.

The independence assumptions implied by Definition 5 have the form:

$$P(x_i | S_i, T_i) = P(x_i | T_i) \quad \text{or, equivalently,} \quad P(x_i, S_i, T_i) \cdot P(T_i) = P(x_i, T_i) \cdot P(S_i, T_i) \tag{11}$$

where $x_i$ is an atomic formula, $S_i = \{s_{i1}, \ldots, s_{ik}\}$ and $T_i = \{t_{i1}, \ldots, t_{il}\}$ are $x_i$'s parents and non-descendants in the primal graph of $\mathcal{L}$. It is important to note that Equation 11 must hold for all truth values of its atomic formulas, namely when each atomic formula $y \in \{\{x_i\} \cup S_i \cup T_i\}$ is replaced by $y$ or $\neg y$. Therefore, it can be encoded by $2^{1+k+l}$ non-linear constraints[5] as follows:

$$(\vec{A}_\alpha \odot \vec{p}) \cdot (\vec{A}_\beta \odot \vec{p}) - (\vec{A}_\gamma \odot \vec{p}) \cdot (\vec{A}_\delta \cdot \vec{p}) = 0 \tag{12}$$

where $k$ and $l$ are the sizes of the parent and non-descendant sets, $\vec{A}_\alpha$, $\vec{A}_\beta$, $\vec{A}_\gamma$ and $\vec{A}_\delta$ are the indicator vectors corresponding to the formulas $\alpha = (x_i \wedge s_{i1} \wedge \cdots \wedge s_{ik} \wedge t_{i1} \wedge \cdots \wedge t_{il})$, $\beta = (t_{i1} \wedge \cdots \wedge t_{il})$, $\gamma = (x_i \wedge t_{i1} \wedge \cdots \wedge t_{il})$, and $\delta = (s_{i1} \wedge \cdots \wedge s_{ik} \wedge t_{i1} \wedge \cdots \wedge t_{il})$, respectively.

The objective $P(f|E)$ is encoded by $\frac{\vec{A}_\omega \odot \vec{p}}{\vec{A}_\epsilon \odot \vec{p}}$, where $\vec{A}_\omega$ and $\vec{A}_\epsilon$ are the indicator vectors of $\omega = (f \wedge e_1 \wedge \cdots \wedge e_t)$ and $\epsilon = (e_1 \wedge \cdots \wedge e_t)$, respectively. We can then obtain the bounds $\underline{P}(f|E)$ and $\overline{P}(f|E)$ by solving min/max $\frac{\vec{A}_\omega \odot \vec{p}}{\vec{A}_\epsilon \odot \vec{p}}$ subject to the constraints defined by Equations 8, 9, 10 and 12, respectively. In our experiments, we used a state-of-the-art non-linear solver such as `ipopt` [31].

---

[5]Some of these constraints are actually redundant and can be removed as suggested by [1]

**Example 3.** *For illustration, consider the LCN $\mathcal{L}$ given by sentences (13)-(17) where $\tau$ is true for all sentences. The primal graph of $\mathcal{L}$ is shown below, where shaded nodes correspond to atoms.*

$$0.6 \leq P\left(a \wedge b\right) \leq 1 \tag{13}$$
$$0 \leq P\left(a \mid c\right) \leq 0.2 \tag{14}$$
$$0 \leq P\left(a \mid \neg c\right) \leq 0.8 \tag{15}$$
$$0 \leq P\left(b \mid d\right) \leq 0.7 \tag{16}$$
$$0 \leq P\left(b \mid \neg d\right) \leq 0.3 \tag{17}$$

*Since $\mathcal{L}$ has 4 atomic formulas $\{a, b, c, d\}$ there are 16 interpretations. For simplicity, we represent their probabilities explicitly as $\{p_{0,0,0,0}, p_{0,0,0,1}, \cdots, p_{1,1,1,1}\}$ where $p_{0,0,0,0}$ is the probability that $a$, $b$, $c$, and $d$ are all false, $p_{0,0,0,1}$ is the probability that $a$, $b$, $c$ are false and $d$ is true, and so on. We enforce $p_{i,j,l,k}$ to be a valid probability distribution by the following constraints:*

$$p_{i,j,k,l} \geq 0, \forall i, j, k, l \in \{0, 1\} \quad and \quad \sum_{i,j,k,l \in \{0,1\}} p_{i,j,k,l} = 1$$

*Sentences (13) and (14), for example, correspond to the following two linear inequality constraints:*

$$\sum_{k,l \in \{0,1\}} p_{1,1,k,l} \geq 0.6 \quad and \quad \sum_{j,l \in \{0,1\}} p_{1,j,1,l} - 0.2 \cdot \sum_{i,j,l \in \{0,1\}} p_{i,j,1,l} \leq 0$$

*Looking at the primal graph, the Markov condition implies that, $c$ and $d$ are independent, $b$ is conditionally independent of $c$ given $\{a, d\}$ and $a$ is conditionally independent of $d$ given $\{b, c\}$, respectively. The latter independence assumption means that $P(a|b, c, d) = P(a|b, c)$ which must hold for all truth values of its atoms. Following the reduction technique suggested by [1], it corresponds to $P(a|b, c, d) = P(a|b, c)$, $P(a|b, \neg c, d) = P(a|b, \neg c)$, $P(a|\neg b, c, d) = P(a|\neg b, c)$ and $P(a|\neg b, \neg c, d) = P(a|\neg b, \neg c)$ which can be encoded by four quadratic equality constraints:*

$$p_{1,1,1,1} \cdot \sum_{i,l \in \{0,1\}} p_{i,1,1,l} - \sum_{i \in \{0,1\}} p_{i,1,1,1} \cdot \sum_{l \in \{0,1\}} p_{1,1,1,l} = 0$$

$$p_{1,1,0,1} \cdot \sum_{i,l \in \{0,1\}} p_{i,1,0,l} - \sum_{i \in \{0,1\}} p_{i,1,0,1} \cdot \sum_{l \in \{0,1\}} p_{1,1,0,l} = 0$$

$$p_{1,0,1,1} \cdot \sum_{i,l \in \{0,1\}} p_{i,0,1,l} - \sum_{i \in \{0,1\}} p_{i,0,1,1} \cdot \sum_{l \in \{0,1\}} p_{1,0,1,l} = 0$$

$$p_{1,0,0,1} \cdot \sum_{i,l \in \{0,1\}} p_{i,0,0,l} - \sum_{i \in \{0,1\}} p_{i,0,0,1} \cdot \sum_{l \in \{0,1\}} p_{1,0,0,l} = 0$$

*The query $P(c)$ corresponds to min/max $\sum_{i,j,l \in \{0,1\}} p_{i,j,1,l}$ and results in the interval $[0, 0.33]$. Similarly, for the query $P(a|b)$, we use the objective $\frac{\sum_{k,l \in \{0,1\}} p_{1,1,k,l}}{\sum_{i,k,l \in \{0,1\}} p_{i,1,k,l}}$ and obtain $[0.85, 1]$.*

**Complexity** The non-linear programs associated with LCNs are non-convex, thus NP-hard to solve in general [25]. One difficulty comes from the large number of quadratic equations involved. The complexity of exact marginal inference in LCNs can be bounded by $O(exp(N))$, where $N$ is the number of interpretations. Therefore, the scalability of exact inference is limited by the size of the LCN. One way to address this issue is to employ state-of-the-art approximate non-linear programming algorithms [2, 32], or to rely on more recent column generation techniques [1]. However, another important direction of our future work is pursuing a belief propagation [26, 19] based scheme for approximate inference in LCNs which will allow us to scale to much larger problems.

## 4   Experiments

We evaluate our proposed approach on random LCNs as well as benchmark problems derived from Mastermind puzzles with uncertainty and a realistic credit card fraud detection application. All our experiments were run on a 2.6GHz CPU with 32GB of RAM.

### 4.1 Random LCNs

For our purpose, we generate random LCNs with $n$ propositional variables $\{x_1, \ldots, x_n\}$ and $n + 3$ sentences of the form $l \leq P(x_i) \leq u$ and $l \leq P(x_i|x_j) \leq u$, where $l, u \in [0, 1]$ and $u - l \geq 0.3$. For each problem size $n$, we generate 10 random instances and for each instance we select 5 different pairs of variables $(x_i \neq x_j)$ to formulate 4 queries per pair: $P(x_i \wedge x_j)$, $P(x_i \wedge \neg x_j)$, $P(\neg x_i \wedge x_j)$ and $P(\neg x_i \wedge \neg x_j)$, respectively. Table 1 reports the average runtime and standard deviation obtained for solving problems with $n \in \{5, 6, \ldots, 12\}$ (200 data points for each value of $n$). We can see that, as expected, the runtime increases exponentially with the problem size due to the complexity of the non-linear programs involved. Therefore, solving random problems with more than 12 variables (over 4096 interpretations) becomes infeasible (i.e., exceeded a 24 hour time limit).

Table 1: Results for random LCNs.

| Size ($n$) | Runtime (sec) |
|---|---|
| 5 | $0.04 \pm 0.03$ |
| 6 | $0.42 \pm 1.34$ |
| 7 | $19.68 \pm 65.2$ |
| 8 | $14.96 \pm 28.9$ |
| 9 | $73.95 \pm 79.2$ |
| 10 | $387.55 \pm 457.1$ |
| 11 | $16794.36 \pm 24852$ |
| 12 | - |

### 4.2 Mastermind Puzzles with Uncertainty

We consider a variant of the popular Mastermind code breaking puzzle [18] in which the code-maker lies randomly at each round. A puzzle with $n$ rounds can be specified by a Bayesian network with 4 hidden code variables $\{h_1, h_2, h_3, h_4\}$, $n$ variables $\{e_1, \ldots, e_n\}$ corresponding to the feedback received in each round and $n$ variables $\{l_1, \ldots, l_n\}$ indicating whether the code maker lied or not when he provided the feedback. We assume that imprecise information about the probability of lying is given by Equations (18)-(21).

Assuming 6 different colors for the hidden code variables (i.e., $c_j \in \{1, \ldots, 6\}, \forall j \in \{1, \ldots, 4\}$) and given the code-maker feedback as evidence, the task is to find the color assignment to the hidden code variables such that its posterior marginal probability interval has the largest upper bound (*maximax*) or, alternatively, the largest lower bound (*maximin*). Namely, we compute:

$$H^* = \underset{c_1, c_2, c_3, c_4}{\operatorname{argmax}} \overline{P}(q|e) \quad \text{and} \quad H^* = \underset{c_1, c_2, c_3, c_4}{\operatorname{argmax}} \underline{P}(q|e)$$

where $q \triangleq q_1 \wedge q_2 \wedge q_3 \wedge q_4$ such that $q_j$ is the color assignment to the $j^{\text{th}}$ code variable $h_j = c_j$, and $e$ encodes the feedback (see the supplementary material for more details).

$$0.3 \leq P(l_i) \leq 0.7, \forall 1 \leq i \leq n \tag{18}$$
$$0.245 \leq P(l_1 \wedge l_2) \leq 0.360 \tag{19}$$
$$0.795 \leq P(l_2 \vee l_3) \leq 0.903 \tag{20}$$
$$0.207 \leq P(l_3 \wedge l_4) \leq 0.273 \tag{21}$$
$$\cdots$$

We evaluate the following competing methods: (i) the Bayesian network (BN) of the puzzle where $P(l_i)$ is assumed to be the midpoint of the probability interval specified by (18); (ii) the credal network (CN) specified by (18); (iii) the ProbLog [28] encoding that use the midpoint $p_{\text{mid}}$ of the probability intervals to annotate the logic formulas (18); (iv) the Markov Logic Network (MLN) representation of (18)-(21) using the midpoints to calculate the weights $w = \log(p_{\text{mid}}/(1 - p_{\text{mid}}))$ of the formulas. We note that BN, CN and ProbLog use only equations (18), while MLN

Table 2: Results for Mastermind puzzles.

| Method | Accuracy | Runtime (sec) |
|---|---|---|
| BN | $65.7\% \pm 2.2\%$ | $0.4 \pm 0.5$ |
| CN (maximax) | $60.3\% \pm 2.3\%$ | $17.0 \pm 0.0$ |
| CN (maximin) | $64.4\% \pm 2.2\%$ | $16.8 \pm 0.6$ |
| ProbLog | $65.7\% \pm 2.2\%$ | $0.4 \pm 0.5$ |
| MLN | $65.3\% \pm 2.1\%$ | $0.5 \pm 0.5$ |
| Nilsson (maximax) | intractable | - |
| Nilsson (maximin) | 0% | - |
| LCN (maximax) | $69.2\% \pm 1.8\%$ | $684.4 \pm 54.1$ |
| LCN (maximin) | $71.1\% \pm 1.9\%$ | $1029.3 \pm 68.9$ |
| LCN (maxent) | **$72.5\% \pm 1.7\%$** | $85.0 \pm 6.3$ |

and LCN are able to utilize logic formulas like (19)–(21). For LCN and CN we use both the *maximax* and *maximin* criteria to determine the hidden code. Furthermore, the LCN based formulation assumes that sentences (19)–(21) are annotated with $\tau = False$ so that they do not imply dependency among the $l_i$ variables. In addition, we consider LCN(*maxent*), an LCN variant that computes a joint distribution of the $l_i$ variables with the largest entropy and assumes that it is the true distribution of lies. For reference, we also ran Nilsson's method [23].

In Table 2 we report the mean accuracy and standard deviation obtained on 10 sets of puzzles, each set using a different random seed and containing 730 random puzzles. We also record the mean total runtime and standard deviation. Accuracy is defined as the fraction of puzzles that an algorithm guessed the ground-truth hidden code correctly. We can see that our proposed LCN approach achieves the highest accuracy compared with its competitors. This is because the LCN is the only method able to aggregate multiple sources of imprecise knowledge in the most effective manner without making any unwarranted independence assumptions among different pieces of knowledge. Although MLN can exploit additional knowledge, it does not gain accuracy because it treats each logic formula as a statistically independent factor and the probabilities as factor potentials; in other words, the probability midpoints are not treated as probabilities in the joint distribution. This is a major weakness of this method, in addition to its inability to handle intervals. In terms of runtime, LCN is slower than its competitors because of the far more complex non-linear programs that need to be solved for each puzzle. However, improving the LCN's runtime is a direction of our future work.

Finally, we note that although Nilsson's method is able to utilize all the logic formulas and their bounds, it does not allow modeling $l_i$ variables as being independent of each other. Given a puzzle, each possible hidden code $c$ corresponds to a truth assignment to the $l_i$ variables; without independence relations, there always exists a joint probability distribution of the $l_i$ variables such that the particular truth assignment has zero probability. Consequently, the lower bound of the posterior probability given a puzzle is zero for any $c$, and therefore the maximin criterion has no basis to make decisions upon. Intuitively, a similar effect should also impair the accuracy of Nilsson(maximax). However, we are unable to verify this empirically: computing the upper bound of the posterior by solving Nilsson's constraint program is computationally prohibitive due to the complexity of the objective function and the size of the program.

### 4.3 Credit Card Fraud Detection

We consider a realistic credit card fraud detection task based on the UCSD-FICO Data Mining Contest dataset [13] which contains 100,000 transactions over a period of 98 days out of which 2,654 are fraudulent. Each transaction is characterized by 16 features, including the transaction amount, timestamp and hashed email address. We used the email addresses as account IDs and split the data into two subsets: one containing 55,750 accounts, each with a single transaction, and another containing 14,374 accounts with multiple transactions, thus totaling 44,250 transactions. These two subsets were subsequently used as training ($T$) and test ($V$) sets, respectively. Additional imprecise knowledge regarding fraudulent transactions and account history is given by the logic rules (22), (23) and (24) which were previously suggested in [21]:

For our purpose, we created 10 randomized tasks such that for each task we sampled half of $T$ and half of $V$, respectively. We then calculated the lower and upper probability bounds of the logic rules (22)-(24) as follows. For each formula and each of the 10 test sets, we measured the conditional probability that the consequent is true given that the antecedent is true. We took the min and max over the test sets and obtain the following probability intervals for the three rules: $[0.65, 0.74]$ for (22), $[0.31, 0.66]$ for (23) and $[0.44, 0.72]$ for (24), respectively.

$$\texttt{Has-FraudHistory}\,(t') \land \texttt{Before}\,(t',t) \rightarrow \texttt{Is-Fraud}\,(t) \tag{22}$$

$$\texttt{Has-ZeroAmountHistory}\,(t') \land \texttt{Before}\,(t',t) \rightarrow \texttt{Is-Fraud}\,(t) \tag{23}$$

$$\texttt{Has-MultiZip}\,(t') \land \texttt{Before}\,(t',t) \rightarrow \texttt{Is-Fraud}\,(t) \tag{24}$$

We evaluated the following methods: (i) a NaiveBayes model learned directly from data; (ii) a Bayesian network (BN) model that expands NaiveBayes by adding the antecedents of the three logic rules as parents of the `Is-Fraud` node and uses a noisy-OR model together with the midpoint of the three probability intervals to represent this conditional probability distribution (the prior

probability of the three new nodes is 0.5); (iii) a credal network (CN) model with the same structure and noisy-OR model as the Bayesian network above but using the three probability intervals as well as the probability interval $[0, 1]$ for the root nodes; (iv) a ProbLog encoding of the NaiveBayes model using the midpoints of the three probability intervals to annotate the ProbLog logic rules; (v) an MLN encoding of the NaiveBayes model extended with the three logic rules and annotated by weights $w = \log(p_{\mathrm{mid}}/(1 - p_{\mathrm{mid}}))$, where $p_{\mathrm{mid}}$ is the midpoint of the respective probability interval.

Table 3 reports the mean and standard deviation of the F1 scores obtained over the 10 random tasks. In this case, the F1 score (defined as the harmonic mean of the precision and recall) is a much better metric than accuracy because the classification task is imbalanced, namely, the probability of fraudulent transactions is much smaller than that of non-fraudulent ones. We also record the mean total runtime and standard deviation. We see again that our proposed LCN method substantially outperforms its competitors in terms of solution quality (F1 score). In

Table 3: Results for fraud detection.

| Method | F1 score | Runtime (sec) |
|---|---|---|
| NaiveBayes | $0.408 \pm 0.110$ | $0.27 \pm 0.01$ |
| BN | $0.090 \pm 0.015$ | $0.29 \pm 0.02$ |
| CN | $0.089 \pm 0.015$ | $0.28 \pm 0.01$ |
| ProbLog | $0.599 \pm 0.048$ | $0.29 \pm 0.02$ |
| MLN | $0.472 \pm 0.094$ | $0.29 \pm 0.01$ |
| Nilsson | intractable | - |
| LCN | $\mathbf{0.630} \pm \mathbf{0.046}$ | $0.29 \pm 0.01$ |

this case, the BN and CN based approaches perform quite poorly because of the unique-assessment requirement. Specifically, the `Is-Fraud` node has three parents and therefore we are no longer allowed to specify $P$ (Is-Fraud). Since $P$ (Is-Fraud) values are measured on the training data (e.g., NaiveBayes), this information is lost in the Bayesian and credal models and, consequently, their false positive predictions increase dramatically. As before, the results demonstrate that LCNs are able to effectively aggregate and exploit multiple sources of information which leads to significantly improved performance. Finally, we see that the performance of Nilsson's method is quite poor in this case as well.

## 5 Related Work

Nilsson's probabilistic logic [23, 24] is perhaps the first system in which the truth values of logical sentences (or formulas) can range between 0 and 1 and are interpreted as the probability of those sentences being true but does not permit specifying independence relations. Bayesian logic (BL) [1] combines probabilistic logic and Bayesian networks in order to capture conditional independence relations among propositions. Markov Logic Networks (MLN) [29] apply the ideas of a Markov network to first-order logic where the weights attached to the logic formulas are used to define a joint probability distribution over all possible interpretations and thus enable uncertain inference. Probabilistic Soft Logic (PSL) [15] combines Markov networks with *soft* or real-valued logic (e.g., Lukasiewicz logic). Probabilistic Logic Programs (PLP) [27] and Stochastic Logic Programs (SLP) [10] are logic programs in which some of the facts are annotated with probabilities. We emphasize that MLN, PSL, PLP, SLP do not allow probability bounds on logic formulas, while BL constrains the formulas to a specific structure. LCN is the only system that addresses these shortcomings.

## 6 Conclusions

In this paper we propose a new probabilistic logic that expresses both probability bounds for propositional and first-order logic formulas with few restrictions and a Markov condition that is similar to Bayesian networks. The formula bounds allow for flexibility in the form and precision of background knowledge that can be utilized, while the Markov condition restricts the space of distributions to enable a meaningful representation of uncertainties. In addition, we show how to perform exact marginal inference to answer queries for the new formalism. Our empirical evaluation on random problems as well as more realistic applications shows promising results, particularly in aggregating multiple sources of imprecise information. Potential future directions include extending to temporal models, further algorithmic innovations to improve the runtime of exact inference, developing efficient approximate inference algorithms and experiments on a wider array of applications.

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
