# OpenReview forum: "Logical Credal Networks"
_NeurIPS.cc/2022/Conference — NeurIPS 2022 Accept_

### Official Review · Reviewer_F77m · 2022-06-26

**Rating:** 7
**Confidence:** 5
**Soundness:** 4 excellent
**Presentation:** 4 excellent
**Contribution:** 2 fair

**Summary:**

Logical Credal Networks (LCN) are an expressive probabilistic logic for expressing credal sets of distributions over interpretations of a logic theory. An LCN is a set of probabilistic statements expressing lower and upper bounds on the probability of a first order logic formula or on the conditional probability of a formula given another formula. The semantics of an LCN is the set of its models where a model is a probability distribution over the interpretations that satisfies the constraints plus the independence constraints that are implied by the model. An LCN can be drawn as a graph and the paper proposes a generalized Markov condition that can be read from the graph, namely that every atomic
formula $x$ is conditionally independent of its non-descendant non-parent variables given its parents.
The paper shows that this Markov condition reduces to the one of Bayesian networks and Markov Logic Networks when the LCN represents one of these special cases.
Marginal inference entails computing upper and lower bounds for $P(f)$ of $P(f|E)$ where $f$ is a formula and $E$ is a set of atomic formulas. The paper proposes a solution to the problem of inference by means of using a non-linear constraint program which has one variable for the probability of each interpretation, where, if there are $n$ atomic formulas, there are $2^n=N$ interpretations. The non-linear program is non-convex so the problem is NP-hard in $N$ and a bound on the complexity is $O(exp(N))$, thus doubly exponential in $n$.

LCNs are evaluated by considering random LCNs, Mastermind puzzles and credit card fraud detection.
On random LCNs, the approach is able to solve problems with up to 11 atomic formulas.
On the Mastermind puzzles, a version of the game is considered where the code maker is allowed to lie. The task is to guess the most probable code set by the code-maker after a number of guesses. LCNs are compared with Bayesian networks, Credal Networks, ProbLog and MLN and found to be more accurate in guessing the correct solution, even if slower.
On the credit card fraud detection problem, a real dataset of credit card transaction is considered and the problem is to predict whether a transaction is fraudulent or not using an LCN composed of three rules together with constraints on them.
LCNs are compared with a Naive Bayes model, a Bayesian Network, a Credal Network, a ProbLog program and an MLN and found to achieve better F1 in a comparable time.

**Questions:**

1) Can you imagine an algorithm that does not have to materialize all interpretations? Or can the language be restricted so that this is not necessary?

**Limitations:**

The limitations of the approach has been presented. The work does not have negative social impacts.

**Strengths And Weaknesses:**

The paper is surely original, to the best of my knowledge this is the first paper applying credal networks to a first order logic language. The expressiveness of the constraints of LCN is very high, giving the possibility of expressing a wide variety of knowledge on the domain, in particular imprecise knowledge and knowledge potentially coming from different sources that, when aggregated, break the acyclicity or unique-assessment requirements of Bayesian and Credal Network.

The paper appears technically sound, correctly identifying a way to extract the implicit independencies in the LCN through the Markov condition that are specifically proposed for LCNs. Importantly, if the LCNs encodes a Bayesian network or an MLN, the proposed Markov condition coincides with that of the specific models. I have only one observation on this:
page 5 says that "the descendants of any atomic node $x$ is always empty" for LCN encoding MLNs. It seems to the that the descendants are always equal to their parents.
The approach for building the non-linear program for inference is correct and I'm convinced that the solution of the program is the solution of the marginal inference problem. I think there is a typo in eq (10) in the second formula:
$A_q$ should be $A_{q\wedge r}$

The experiments are sufficiently extensive to demonstrate the usefulness of the added expressiveness of LCNs with respect to the main competing methods. The random LCNs experiments give a clear idea of the scalability of the approach (11 variables), while the Mastermind and credit card fraud experiments show the improved solution quality of LCNs.

The main weakness of the approach is the scalability: 11 atomic formulas are not many for a relational domain. This is due to the fact that the whole set of interpretations of the first order logic theory has to be considered in the inference algorithm, making it doubly exponential in $n$, the number of atomic formulas. This is in stark contrast with Bayesian networks and MLNs where inference avoids generating all possible interpretations and this is the main strength of these approaches. It would be a very interesting direction for future work to overcome this limitation.

Overall the paper is very clearly written, expressing the concepts in details and with illuminating examples. The only thing not very clear is the list of equations (18)-(21) that contain dots, meaning that there are others, but then the text only refers to (19)-(21), leading one to think there are only those. The situation is clarified in the supplementary material where the exact procedure for generating those equation is explained, confirming that there are many more.
This point could be clarified.

Minor comments:

Page 6, last line $P(a|b,\neg c$ is missing a closing parenthesis

---

> ### Author Response · Authors · 2022-07-31
> **Response to Reviewer F77m**
>
> Thank you for your comments. We will fix all the presentation issues found.
>
> We agree that the main limitation of our approach is scalability. Clearly, using and/or adapting lifted inference ideas to LCNs as you suggested is one way to speed up inference in LCNs. This is actually part of our current research effort in this area.
>
> However, since the NeurIPS submission, we developed a novel iterative message-passing scheme for approximate inference in LCNs. It is essentially an algorithm that propagates messages along the edges of a factor graph associated with the input LCN. These messages involve solving much smaller local non-linear constraint programs. In our experiments, we reduced the running time on random LCNs with 11 variables to around 20 seconds on average. This essentially allows us to scale to much larger LCNs involving hundreds or even thousands of variables. We will include a discussion on our approximation scheme for LCNs.

---

> > ### Comment · Reviewer_F77m · 2022-08-04
> > **inference**
> >
> > Thanks for the comments. Yes, please include a discussion of approximate inference in the paper.
> > I will raise my evaluation to Accept

---

### Official Review · Reviewer_ak6q · 2022-07-09

**Rating:** 7
**Confidence:** 3
**Soundness:** 3 good
**Presentation:** 3 good
**Contribution:** 3 good

**Summary:**

This paper proposes a new graphical model and language to express probabilistic constraints over logical formulas, either in the propositional language or in first order logic, expressed either as unconditional or conditional formulas. Compared to previous languages, it offers more flexibility in terms of modelling, and assume variable independence as a default assumption, rather than as a declared property. Dependence then has to be explicitly stated. The paper introduces Markov conditions for this kind of models, and demonstrates through some small-scale experiments that the model increased expressiveness allow one to increase inference quality.

**Questions:**

* In the paper, FOL problems are solved by grounding them, which clearly induces a quick explosion if the number of variables increase. Is this grounding actually the only way to solve the problem or could solvers directly adapted to FOL problems be used, e.g., by considering fragments of FOL rather than all of it? Echoing to this question, my feeling is that here authors consider that any Boolean sentence can be provided in each parts of the formulas. Would the inference problem become simpler if one were to restrict the possible formulas to peculiar fragments of logic (still expressive enough to warrant application, but with properties allowing for faster inferences).

* Any ideas about approximation schemes for this particular language that would allow for some scalability? In the paper, this is left for future work, but maybe the authors have some ideas about it they can share. For example, would linearisation schemes help and allow to use LP tools?

* In the comparisons with other approaches, authors mainly consider other probabilistic graphical models, at the exception of problog. What about using probabilistic versions of answer set programming, or PSAT solvers?

Typos or other small remarks:

* L195 and others: reference to equation is not between parenthesis. This is unusual (but maybe voluntary)

* L217: missing parenthesis after $P(a|b,\not c$

**Limitations:**

The main limitation of the presented approach, i.e., its scalability, is mentioned in the paper.

**Strengths And Weaknesses:**

**Originality**

+: this is a new model to express probabilistic logic constraints, be they precise or imprecise. The proposal is quite convincing.

**Quality**

+: the paper is quite well-written and quite understandable. The proposal is sound, and the first results seem promising, even if scalability of the proposed will clearly be an issue.

**Clarity**

+: the paper presents the model clearly, even for a non-expert.

**Significance**

+: a new model that seems to present quite interesting properties in terms of logical inference (e.g., allowing for bounds over probabilistic information while not making inference quickly vacuous), and present a quite expressive model.

-: two limits of the paper are clearly on the one hand its scalability if one has to perform exact inferences over it, and the need (?, see questions) to ground FOL problems into propositional ones.

---

> ### Author Response · Authors · 2022-07-31
> **Response to Reviewer ak6q**
>
> Thank you for your comments. We agree that the main limitation of our approach is scalability. However, since the NeurIPS submission, we developed an iterative message-passing scheme for approximate inference in LCNs. It is essentially an algorithm that propagates messages along the edges of a factor graph associated with the input LCN. These messages involve solving much smaller local non-linear constraint programs. In our experiments, we reduced the running time on random LCNs with 11 variables to around 20 seconds on average. This essentially allows us to scale to much larger LCNs involving hundreds or even thousands of variables. We will include a discussion on our approximation scheme for LCNs.
>
> In our current approach we did not impose any restrictions on the logical formulas (either propositional or FOL) so that we wouldn’t limit expressivity. We agree that it is equally important to consider simpler fragments of FOL to speed up inference without limiting expressivity. This is currently part of our ongoing research effort in this area.
>
> We will fix the typos and references to equations.

---

### Official Review · Reviewer_WFaC · 2022-07-11

**Rating:** 7
**Confidence:** 4
**Soundness:** 3 good
**Presentation:** 3 good
**Contribution:** 3 good

**Summary:**

The paper proposes a new probabilistic logic based on interval-valued probabilistic statements. This can be seen as a simultaneous generalisation of both Markov Logic Network and Credal Networks that, in turn, extend Bayesian networks. The syntax and semantics are provided and the inference is shown to correspond to a non-linear (hard) task. The experiments (Mastermind with Uncertainty, Fraud Detection Data) show promising results as MLN, BN, CN and ProbLog approaches are outperformed by the proposed technique.

**Questions:**

- In proving the hardness of LCN the authors follow the mapping to non-linear programs. What about using the link with complexity results for CN (see e.g., Mauà et al., JAIR 2012)? As LCN extends BN, it should also (trivially) extend CN inference.
- It is not clear to me whether or not a "grounding" operations allows to write a LCN as a (larger) CN. This would be relevant for inference.
- I consider the "Remark" paragraph quite important. The fact the LCN extendes both MLN and BN/CN is a strong justification for this new probabilistic logic. What about making the result more formal (i.e., a proposition with a proof)? I see the derivation is almost straightforward, but making this even more clear would be potentially important.
- I am very happy with the discussion in S3.3, but I miss at least an extensive comments on how the existing literature (mostly in the CN area) addressed similar (multilinear or fractional multilinear) optimisation tasks. I am quite sure that some of the techniques considered for CN could be adapted to LCN and an extensive discussion on that point would be important.
- I am not sure whether, in the experiments, MLN is equivalent to LCN with sharp probabilistic assessments in the midpoints. If this is not the case also that setup would be interesting.
- The above point is also related to the fact that, in the experiments maximin/maximax/maxent, are clearly reasonable approaches to decision making with imprecise information if the goal is to output a single option, but it might be also interesting to see what happens if multiple outputs are considered. The simplest experiment would be to compare the maximin and the minimax solutions and to evaluate separately the accuracy (of some other, precise approach) when these two solutions are equal and when they are different. We might expect a clear separation between the two values.
- I am a bit surprised by the very similar execution times in Tab3. Is there an explanation for that?
- Regarding loopy propagation, Zaffalon's 2U algorithm for CN allows for polynomial inference in tree-shaped Boolean CNs. Cozman et al. provided a loopy version of 2U. These techniques can be probably applied to LCN. Of course this is also related to the work of Polastro & Cozman properly cited by the authors. What about commenting also this direction?



**Limitations:**

I don't see urgent issues related to the societal impact. Regarding the limitations, these are mostly related to the computational complexity, and the authors are very explicit on that.

**Strengths And Weaknesses:**

Strengths
- The contribution is clearly novel and, in a sense, is filling a gap in the literature (Probabilistic Logic in interval-valued statements).
- Although mostly heuristic, I don't see issues with the technical derivation of the results (e.g., the fact that LCN generalises both MLN and BN).
- The presentation is clear and quite accessible.
- The experiments are clearly showing the advantages of the proposed approach.

Weaknesses
- My only concern with the paper is related to the inference part. The authors clearly describe how to solve an LCN inference with a generic non-linear solver, but no particular attention is paid to exploit the specificities of the task in order to find faster solutions.

---

> ### Author Response · Authors · 2022-07-31
> **Response to Reviewer WFaC**
>
> Thank you for your comments. We appreciate the reviewer’s suggestion to make stronger connections with the complexity results and inference algorithms for CNs. However, these results might not be easily extended to LCNs because LCNs don’t have to be acyclic and they also break the unique-assessment assumption.
>
> Propagation algorithms like L2U/2U/IPE developed for CNs aren’t compatible with LCNs in general, unless the given input LCN encodes a CN. The following simple example LCN shows that L2U/2U computes incorrect results: 0.2 <= P(a) <= 0.3; 0.6 <= P(b|a) <= 0.7; 0.1 <= P(b|!a) <= 0.2; 0.3 <= P(b) <= 0.4. If we query P(b), the correct answer is [0.3, 0.35] whereas 2U/L2U gives [0.1, 0.26] which is incorrect.
>
> However, since the NeurIPS submission, we developed a novel approximation scheme for LCNs. It is essentially an iterative message-passing algorithm that propagates messages along the edges of a factor graph associated with the input LCN, where messages involve solving smaller local non-linear constraint programs. In our experiments we reduced the running time on random LCNs with 11 variables to a little over 20 seconds which allows us to scale to much larger problems (with hundreds or even thousands of variables). We’ll include a discussion on the approximation scheme.
>
> We’ll revise the Remark paragraph and include a formal result regarding the generalization of BNs and CNs.
>
> Regarding the results in Table 3, the resulting LCNs are relatively small and the corresponding constraint programs are very easy to solve by ipopt. This is the reason for similar running times with the other methods.
>
> In our Mastermind experiments we are essentially solving a Marginal MAP task in a brute-force manner. We appreciate your suggestion to extend the experiment to a top k MMAP task and we will consider it in our future work.

---

### Official Review · Reviewer_Lye7 · 2022-07-11

**Rating:** 4
**Confidence:** 4
**Soundness:** 2 fair
**Presentation:** 2 fair
**Contribution:** 2 fair

**Summary:**

A new probabilistic model called Logical Credence networks is proposed. The model allows us to specify ranges of probabilities for logical rules. The probabilities may be specified as conditionals or marginals. A mapping from the logical structure to a graphical structure is proposed that can specify independencies. An inference procedure is developed that is based on solving a system of equations over the grounded logical theory. This gives us an upper and lower bound on the marginal probabilities.

Experiments are performed over synthetic structures as well as mastermind puzzles and a fraud detection dataset.

**Questions:**

To make the model complete, I think there needs to be a clear specification of how we can learn such models, so a discussion on this will be useful.
Also, it would be useful to hear about some of the concerns in the previous section.

**Limitations:**

Limitations (e.g. scalability) has been discussed.

**Strengths And Weaknesses:**

Strengths

1. The ability to specify ranges or bounds for the formulas may be useful instead of a single parameter (e.g. in MLNs)
2. Constraining the marginal probabilities based on the independencies may result in probabilities that are more intuitive.

Weakness
1. The clarity of the contribution is somewhat lacking. For example,  the  representation and the inference procedure is presented, but how are the parameters learned? In the experiments, it seems to be done based on the test data (e.g. the fraud detection task). Without a standard learning procedure, it seems the model is a bit incomplete.
2. From a practical perspective, the model seems to be not as scalable and am not sure if it would apply to real-world settings.
3. I am not sure about the comparison with approaches such as MLNs and Problog since the weights are not learned here as I see it. Also, the statements in lines 280, 281 may not be fully accurate. In MLNs, the formulas are not independent since they are connected through a factored Probabilistic  Graphical Model. Maybe this needs to be rephrased.
4. From a higher-level perspective, there are other types of models that seem similar. The one that comes to mind is PSL (Probabilistic Soft Logic) that allows arbitrary function to be specified over logical formulas (the ranges may be a special case of this). A discussion on the significance of the proposed approach will be useful.

---

> ### Author Response · Authors · 2022-07-31
> **Response to Reviewer Lye7**
>
> Thank you for your comments. We would like to emphasise that the main focus of our paper was to formally introduce the LCNs and to show how to perform exact inference in LCNs. Clearly, the underlying assumption is that the logical formulas and the probability bounds are elicited from the domain experts. However, learning LCNs from data, both the structure and parameters, is equally important and is part of our current research effort.
>
> We’ve actually tackled the scalability issue and, since the NeurIPS submission, we developed an approximation scheme for LCNs. It is essentially an iterative message-passing algorithm that propagates messages along the edges of a factor graph associated with an LCN. In our experiments we managed to reduce the inference time to a little over 20 seconds on random LCNs with 11 variables and therefore we can now scale to much larger LCNs with hundreds or even thousands of variables. We’ll include a discussion on approximate inference for LCNs.
>
> We appreciate the connection with PSL and we’ll expand the discussion in the related work section. As far as we know, PSL relies on Lukasiewicz’s real-valued logic whereas LCN uses classical logical operators. Moreover, it’s not very clear how to extend PSL with bounds on the weights associated with the logical formulas.

---

### Meta-Review · Area_Chair_Dz6o · 2022-08-23

**Recommendation:** Accept
**Confidence:** Less certain

**Metareview:**

The theoretical connection of logic and probability via credal networks has been seen as positive. The computational complexity is a negative point (as it is even harder than credal networks, which are already very hard to solve). The expressive representation framework has outweighed the high complexity in the discussions. There are many suggestions that could be integrate into the manuscript, including with other works on credal networks, decision criteria and maximum entropy. Approximate methods may well be the only way forward to scale it, and theoretical properties can be further expanded clarified in terms of algorithmic complexity and representation power, not to mention how to learn from data plus constraints (hinting here to future work, obviously). It is a huge area with many open problems, and the paper has been considered as opening some doors (including the connection to MLN). The recommendation is not unanimous though, and I reckon that a credal set would better represent the opinion of the experts about this submission.


**Award:**

No

---

### Decision · Program_Chairs · 2022-09-14

Accept